# The Relationship Between Body, Mind, and Social Characteristics in a Sample of a Training Program for Developing Social and Personal Resources: A Network Analysis

**DOI:** 10.3390/ijerph21121654

**Published:** 2024-12-11

**Authors:** Christoph Janka, Maike Höcker, Thomas E. Dorner

**Affiliations:** 1Center for Public Health, Department of Social and Preventive Medicine, Medical University of Vienna, Kinderspitalgasse 15/1, 1090 Vienna, Austria; 2Faculty of Medicine, University of Duisburg-Essen, Hufelandstraße 55, 45147 Essen, Germany; 3Karl-Landsteiner Institute for Health Promotion Research, Haus der Barmherzigkeit—Clementinum, Paltram 12, 3062 Kirchstetten, Austria; 4Academy for Ageing Research, Haus der Barmherzigkeit, Seeböckgasse 30a, 1160 Vienna, Austria

**Keywords:** network analysis, network comparison test, training, personal growth, transcendence, wellbeing, life satisfaction, sense of coherence, social relationships, meaning in life

## Abstract

In this cross-sectional study, the interactions between demographic characteristics and the body, mind, and social dimensions among participants of a holistic training program for social and personal development were examined and compared to a control group. The sample involved 223 adults from Austria, Germany, and Switzerland, with 80 (37.2%) participants in the training group already having finished the training. To explore the variable relationships and compare group structures, advanced network analyses and a network comparison test were applied. The network analysis identified the training as a central variable, being linked to aspirations for societal impact and personal growth (0.31), aspirations for the pursuit of fame and wealth (0.29), transcendence (0.19), the desire to improve the quality of relationships (0.19), aspirations for personal wellbeing and relationships (0.15), and the presence of social resources (0.13). The group comparison revealed that the control group primarily connected through the “Sense of Coherence—Meaning in Life” axis, whereas the training group showed more complex linkages involving wellbeing, life satisfaction, meaning in life, and transcendence, underlining the training program’s positive effect on personal growth, societal impact aspirations, and transcendence. Overall, the network comparison test revealed significant differences in network structure and overall connectivity between the training and the control group.

## 1. Introduction

### 1.1. Background

The rapid transformations and challenges of our modern life put considerable pressure on mental health and overall wellbeing. The World Health Organization (WHO) recognizes depression as a major global disability, affecting 264 million people worldwide, and anxiety disorders impacting approximately 284 million individuals [1]. Economic factors further worsen mental health issues; the European Union reports a financial burden of more than EUR 600 billion due to stress-related mental illnesses within the working population [2]. Additionally, for a large portion of the adult population, the impacts of the COVID-19 pandemic have further increased stress levels [3].

Social issues such as loneliness have become significant public health concerns as studies suggest that these conditions are linked with an increased risk of major health problems, including heart disease, stroke, dementia, and premature mortality [4]. Economic downturns have been found to worsen symptoms of anxiety and depression, which makes mental health issues even more complicated [5]. And even when digital connectivity has increased, it often contributes to greater feelings of isolation and anxiety [6].

Research underscores the protective role of developing personal and social resources to improve resilience and enhance wellbeing [7,8]. Social support, for instance, has been shown to be a buffer against psychological stresses caused by rapid societal changes [9]. Strong social relationships have been linked with improved health outcomes and increased longevity [10,11], as well as lifelong happiness and health, showing the importance of social connections [12].

While these findings highlight the importance of personal and social resources for mental health, a critical need arises with them: proactive interventions for the general non-clinical population, which has shown to experience mental health issues to a notable degree as well [13]. Such interventions can help individuals develop and strengthen these vital resources before mental health issues arise, potentially preventing the onset of clinical conditions and promoting overall wellbeing. For example, mindfulness-based stress reduction programs have been shown to reduce symptoms of anxiety and depression in both clinical and non-clinical populations [14]. Cognitive-behavioral training interventions have also demonstrated efficacy in enhancing coping strategies and reducing stress levels [15].

Despite the known benefits, many current intervention programs remain focused on single dimensions and often do not integrate personal growth with social integration effectively [16]. This separation can significantly reduce the overall effectiveness of programs, as the interplay between these dimensions, such as supportive networks, community engagement, and effective communication skills, is critical for managing the stresses of daily life and achieving a balanced lifestyle [7,9].

Also, while there are numerous interventions tailored for clinical patients, there is a significant gap in comprehensive promotion programs designed for the general population. [17,18]. Promotion and preventative measures are crucial as they can mitigate the onset of mental health issues by enhancing resilience and fostering wellbeing before problems arise. In response, holistic training programs that simultaneously address personal development and social integration to promote wellbeing are getting more into focus [19]. Previous studies have highlighted the potential of holistic interventions in improving mental health outcomes, enhancing life satisfaction, and strengthening social bonds [20,21,22].

In this study, a holistic intervention program, designed to improve participants’ mental health and resilience and strengthen social bonds, was evaluated. This 1-year program addressed key topics such as stress and somatic symptom management, life satisfaction, social relationships, health behavior, and overall wellbeing and aimed to equip participants with practical skills integrated into a comprehensive framework for daily application. This should potentially improve the effectiveness of such interventions and inform future program designs and policymaking.

By addressing critical gaps in existing models, this approach aimed to refine theoretical frameworks for personal and social development and offer insights for training providers, policymakers, and mental health professionals that may help enhance individual and community wellbeing.

### 1.2. Aims and Objectives

It was the aim of the study to investigate the interactions between demographic factors and the body, mind, and social characteristics of participants within the training program. It also aimed to compare the network structures of the training and control groups to find unique and common patterns of interaction and explore their modifications between the two groups. Finally, it aimed to determine the most influential variables within these networks and analyze how their impact varies between participants who have taken the training and those who have not.

## 2. Materials and Methods

### 2.1. Study Design

This study employed a cross-sectional design to investigate the interaction between demographics and body, mind, and social characteristics in participants of a training program aimed at developing social and personal resources. The reason for the cross-sectional design was due to practical constraints—training participants had previously taken and finished the training, making a longitudinal design with baseline measurements infeasible. This approach was chosen to better reflect real-world conditions where individuals self-select into training programs. To mitigate the effects of biases and enhance the validity of the findings, a control group was recruited and statistical controls for potential confounders were applied. In the chosen network analysis model, all variables have been controlled against each other using elastic-net regularized neighborhood regression.

### 2.2. Intervention

The Resonanz© Practitioner intervention, developed and conducted by Institute Kutschera, is a comprehensive 1-year training program for adults, designed to improve personal and social resources and enhance participants’ overall quality of life [23]. By equipping individuals with practical tools and knowledge, the program aims to create sustainable positive changes in various aspects of their lives, including mental health, relationships, and personal development.

The program included six training modules (five 3-day modules and one 5-day module), including theoretical knowledge and practical exercises. Participants attended 10 in-person one-on-one sessions with trainers to review progress and participated in 30 h of peer group meetings between modules. The trainings were conducted in a face-to-face setting in Austria, Germany, and Switzerland.

The training was based on Lozanov’s superlearning [24] and positive psychology principles [25] and aimed to create a relaxing and supportive learning atmosphere. Exercises to foster individual strengths had a focus on personal growth and followed a specific protocol designed by the training provider and applied presentation methods that appealed to visual, auditory, and tactile perceptions [23]. The training covered several key topics, see Table 1.

The participants were encouraged to actively engage in the activities to maximize the benefits of the training.

### 2.3. Participants

Adults from Austria, Germany, and Switzerland—the training provider’s target countries—participated in the study; recruitment and data collection were conducted in April and May 2023. The recruitment included an outreach in the community and advertisements on platforms such as Facebook, Instagram, and academic and professional knowledge exchange online forums. Participants of the training group were recruited directly among training alumni through the training provider.

The eligibility criteria included being between 18 and 70 years old and being fluent in German, as the trainings were held in German. Additional criteria for the training group included having completed at least one training program. The exclusion criteria were requiring less than half of the expected time (40 min) to finish the survey or answering the control questions incorrectly.

Informed consent to participate in the study and to publish the results of this study was obtained from all participants via a form integrated into the survey prior to their inclusion in the study. The process followed a standardized protocol, and the text was provided by the ethics commission of the Medical University of Vienna. The study was conducted in accordance with the Declaration of Helsinki and approved by the Ethical Committee of the Medical University Vienna (Ref: 1592/2019) on 25 October 2019. The study protocol was registered at clinicaltrials.gov (Identifier: NCT04165473).

### 2.4. Data Collection Instruments

#### 2.4.1. Online Survey Tool

Data were collected via an online questionnaire using the Sosci-Survey platform [26]. Sosci-Survey is a robust tool for academic research, which ensures an efficient and streamlined online survey process and guarantees data integrity and security. The platform was configured to prevent the submission of incomplete surveys, thereby ensuring no missing data.

#### 2.4.2. Questionnaire Domains and Instruments

The survey included standardized questionnaires and custom items designed to measure the body, mind, and social characteristics of the participants. Figure 1 describes the domains body, mind, and society, and the questionnaires assigned to each domain. Further details were presented earlier in the published study protocol [23].

The full list of instruments is shown in Table 2 and has been described in detail in the study protocol [23].

### 2.5. Statistical Analysis

Initially, participants who did not meet the eligibility criteria were excluded. The remaining data were then assessed for plausibility using descriptive statistical analysis, and any implausible values were verified against the original participant records. IBM^®^ SPSS^®^ Statistics for Mac (Version 29; SPSS, Inc., Chicago, IL, USA), JASP 0.18.3 (Jasp Team 2024), and a variety of R packages in R 4.4.3 (R Foundation for Statistical Computing, Vienna, Austria) were used for the statistical analyses. SPSS was used for general statistical analysis, JASP for network analysis and visualization, and R for advanced network comparison.

To identify variables relevant to the network, both parametric and non-parametric methods were employed. T-tests, U-tests, and chi-square (or Fisher Exact test) tests were used to determine significant differences between the control and training groups. All tests were two-sided, with a significance level set at *p* < 0.05.

For variables with numerous related subscales, explorative factor analysis (EFA) was used to detect underlying relationships and simplify the data for network analysis [38]. Different subscales, measured on varying scales, were z-standardized to normalize them.

Only variables with scales and subscales that met specific quality criteria were considered for further analysis. Factorial validity was assessed using principal component analysis with Varimax rotation. The thresholds applied for each variable were Kaiser–Meyer–Olkin (KMO) > 0.6 [39], Bartlett’s test of sphericity *p* < 0.05 [40], Eigenvalues > 1 [41], Scree-plot criteria [42], communalities > 0.3 [43], and Measures of Sampling Adequacy (MSA) > 0.7 [44]. Reliability was ensured by requiring inter-item correlations > 0.5 [45], item-total correlations > 0.5 [46], and Cronbach’s alpha > 0.6 [47] to measure internal consistency.

Values for continuous normally distributed variables are presented as the mean ± standard deviation (SD). For continuous variables that are not normally distributed, values are shown as the median, using the rank mean method, which averages ranks assigned within the dataset to provide a central tendency measure. Nominal variables are expressed as counts (N) and percentages (%) to illustrate frequency distribution within the group. For ordinal variables, values are also presented using the median and the rank mean, summarizing central tendency by averaging ranks assigned to ordered, but not equally spaced, categories.

To analyze the network structure, a Mixed Graphical Model (MGM) was employed, which allowed the inclusion of continuous, ordinal, and nominal variables. The model was estimated using the “mgm”, “qgraph”, and “bootnet” R-packages via JASP.

Two distinct network analyses were used in this study. The “Unified Network Analysis” involved all participants in a single network model, including training status as a sociodemographic variable. The “Comparative Network Analysis” compared the network structures of the control and training groups separately to identify unique and common interaction patterns.

In the network, edges between nodes describe the pairwise interactions between these variables, conditioned on all other nodes. Each edge in the network represents either a positive regularized association (green edges) or a negative regularized association (red edges). Edges involving a categorical variable are kept in grey. Ordinal variables were coded as squares and nominal variables as triangles and continuous variables were represented as circles. Only edge weights > 0.1 were displayed, as weights below this threshold were considered too weak to provide meaningful insights [48].

Cross-validation (CV) was employed to provide a reliable estimate of model performance, to prevent overfitting, and to ensure that the model generalizes well to new data [49]. The calculation was run with 10 folds and using 1000 non-parametric bootstraps [50,51]. The AND rule was applied to ensure that both regressions between nodes were significant.

To identify the most influential variables within the network, centrality indices were calculated. Betweenness Centrality describes nodes that acted as bridges, Closeness Centrality measures how quickly a node can reach all other nodes, and Strength Centrality reflects its overall influence level within the network.

The network comparison test (NCT) compared the network structures between the groups. The network structure invariance test compared the distributions of edge weights. The global strength invariance test compared the overall connectivity through the weighted absolute sum of edge weights between two networks [52,53].

## 3. Results

### 3.1. Descriptive Statistics and Primary Analysis Results

As shown in Table 3, a total of 223 participants were included in the study, with 83 participants in the training group and 140 participants in the control group. Participants were recruited from Austria (*n* = 191), Germany (*n* = 26), and Switzerland (*n* = 6). The median age was higher in the training group compared to the control group. Both groups had a higher proportion of females.

The majority of participants in both groups mentioned secondary school or higher education as their highest education. Whereas, in the control group, secondary school ranked highest, and it was university or college in the training group.

A higher proportion of participants in the training group reported household incomes above the national median compared to the control group, and a similar trend was found for individual income. Full-time employment was more common in the control group, while self-employment was more prevalent in the training group. Regarding the duration of the recent employment, both groups had a similar median duration.

Most participants in both groups lived in urban areas. The majority of participants in both groups lived with their own families and stated they were married or partnered. Compared to the control group, more participants in the training group had children.

In summary, participants in the training group tended to be older, more likely to be female, have a higher household income, and be self-employed, while the control group had a higher proportion of full-time employed participants.

### 3.2. Selection of Variables for the Network Analysis

#### 3.2.1. Sociodemographic Variables

The following variables met the criteria described in Section 2.5 and were included in the network analysis—Table 3 provides a detailed analysis of age, sex, education, household income, employment, and family status.

#### 3.2.2. Body, Mind, and Social Characteristic-Related Variables

The following variables met the criteria described in Section 2.5 and were included in the network analysis—Table 4 provides a detailed analysis of perceived stress level, physical condition at wake-up, contribution to satisfaction and wellbeing, physical activity, satisfaction with life, quality of social relationships including workplace environment and private environment, desire to improve quality of relationships, psychological wellbeing, aspiration factors such as personal wellbeing and relationships, pursuit of fame and wealth, societal impact and personal growth, presence of meaning in life, sense of coherence, resource inventory including social resources, autonomy, and transcendence.

### 3.3. Network Analysis

The network analysis was conducted to investigate the complex interactions between various demographic, body, mind, and social variables in all participants, with training status (a dichotomous variable with the control group not having taken a training, while the training group has) included as a variable. Only those variables that demonstrated a significant difference between the training group and the control group (Table 3 and Table 4) were included in the analysis. The continuous variables selected as nodes were perceived stress, QoSR—workplace environment, QoSR—private environment, psychological wellbeing, physical activity, physical condition at wake-up (i), life satisfaction (i), contribution of life aspects to satisfaction and wellbeing, sense of coherence, autonomy, social resources, transcendence, aspirations (personal wellbeing and relationships, societal impact and personal growth, pursuit of fame and wealth), presence of meaning in life, age, household income, sex, and family status. Dichotomous nominal variables were coded as continuous/metric. Additionally, the desire to improve quality of relationships was treated as an ordinal variable, while education and employment were considered nominal variables. The variables physical condition at wake-up and life satisfaction, denoted with an ‘(i)’ in their labels, have been inverted so that higher values represent better physical condition upon waking and greater life satisfaction, respectively.

### 3.4. Unified Network Diagram

The network diagram for all participants in Figure 2 illustrates the relationships between the selected variables, including training status. Variables were color-coded as follows: “Sociodemographic” variables (green), “Training Aim” (blue), and “Outcome” variables (orange).

Table 5 highlights the edge weights of the strongest connections with absolute weights > 0.2 between various variables, which shows the relationships between training aim variables and outcome variables with edge weights > 0.1 and presents how training status relates to other factors, with edge weights > 0.1. The full dataset of edge weights is detailed in Appendix A.

Training status and age had the strongest positive correlation. Training status also showed significant positive associations with societal impact and personal growth, pursuit of fame and wealth, transcendence, desire to improve quality of relationships, education, social resources, and the aspiration for personal wellbeing and relationships. Conversely, a negative association was found between training status and sex. Physical condition at wake-Up (i) had a positive correlation with psychological wellbeing. Social resources showed a positive correlation with personal wellbeing and relationships, while aspirations for societal impact and personal growth were negatively correlated with aspirations for personal wellbeing and relationships.

Sense of coherence, the training aim variable being most related to outcome variables, showed significant connections with four outcome variables: QoSR—workplace environment, perceived stress, and physical condition at wake-up (i), and life satisfaction (i). Social resources had strong positive correlations with QoSR—Private Environment and Contributions of Life Aspects to Satisfaction and Wellbeing. Aspirations for personal wellbeing and relationships was positively correlated with QoSR—private environment and contributions of life aspects to satisfaction and wellbeing.

### 3.5. Centrality Indices

Figure 3 shows the z-standardized centrality values for betweenness, closeness, and strength for each node. Node 24, “Training Status”, had the highest z-value in all three centrality indices, which demonstrates its critical role and influence within the network. Node 5 and node 22, “Physical Activity” and “Employment”, had the lowest z-values in all three indices, indicating their minimal role in the network’s overall connectivity and information flow. Full details are presented in Appendix A.

### 3.6. Comparison Between Training and Control Group Networks

For the comparison of the network structures between the control and training groups, “Training Status” was used as a split variable. Figure 4 shows the network diagrams for the control group and the training group, respectively.

Table 6 highlights the strongest connections, selected based on their absolute edge weights between various variables in the control group and in the training group network. Full details are presented in Appendix A as well as in Appendix A.

In Table 7, a list of variables is presented for both the control and training groups. The median number of the variables’ connections in each group was chosen as the lower limit for inclusion in the list. The variables are ranked based on the number (#) of connections they have with absolute edge weights greater than 0.1. When they have the same number of connections, the variables are further ranked by the sum of the absolute edge weights of their connections. This identified the main aspects within each group and helped to find similarities and differences between the control and training groups.

Control Group: Showed a diverse range of variables with significant connectivity: “Aspirations: Personal Wellbeing and Relationships” is on top of the list with eight connections, followed by “QoSR—Private Environment” with 7, and “Presence of Meaning in Life” and “Aspirations: Societal Impact and Personal Growth”, each with 6 connections. Further “Sense of Coherence”, “Aspirations: Pursuit of Fame and Wealth”, “Psychological Wellbeing”, “Life Satisfaction (i)”, and “Perceived Stress”, all had five connections.

Training Group: “Autonomy” had the most connections (6), followed by “QoSR—Private Environment”, “Presence of Meaning in Life”, “Life Satisfaction (i)”, “Desire to improve Quality of Relationships”, “Sense of Coherence”, and “Perceived Stress” with four connections each. “Psychological Wellbeing”, “QoSR—Workplace Environment”, and “Age” were also notable with three connections each.

### 3.7. Network Comparison Test (NCT) Results

The NCT analysis compared the network structures between the control and training groups and revealed significant differences in network structure invariance (M = 0.35, *p* = 0.004) and global strength invariance (S = 5.80, *p* = 0.013).

## 4. Discussion

Our study successfully addressed its three primary aims by examining demographic patterns among participants, comparing network structures between groups, and identifying influential variables within these networks, revealing several notable patterns in the relationships between training participation and psychosocial characteristics.

The first aim was to investigate the interactions between demographic factors and the body, mind, and social characteristics within the training program:

Examination of the demographic characteristics revealed distinct patterns between training and control groups, with key differences in age, gender distribution, and socioeconomic factors. The training group was older and had a higher proportion of females compared to the control group. This aligns with previous research suggesting that women are more likely to participate in personal development programs [54]. Additionally, older adults may seek out such training programs as they prioritize meaningful activities and social engagement in later life, consistent with the socioemotional selectivity theory [55,56]. Education and income levels were higher in the training group, indicating that individuals with more resources and educational achievement are more likely to engage in personal development activities, which has also been shown in earlier research [57,58]. Being married or partnered and having children were also more common among training participants, possibly reflecting a desire to enhance family relationships [59]. Furthermore, self-employment was more prevalent in the training group, which may reflect the need for personal growth and autonomy, important traits for success as an entrepreneur [60]. These findings help to understand more about the determinants of individuals open to attending this kind of training.

The second aim was to compare the network structures of the training and control groups to find unique and common patterns of interaction and explore their modifications between the two groups:

Network analysis of psychosocial characteristics revealed differential patterns of connections between the training and control groups. The training group exhibited stronger connections between Life Satisfaction, Meaning In Life, Transcendence, And Psychological Wellbeing. These results are in line with research that found that the presence of meaning in life is strongly associated with Life Satisfaction and other positive outcomes [61].

Autonomy, besides social relations, was emphasized in the training group’s network structure, indicating that the training group showed higher self-determination. According to the Self-Determination Theory, autonomy is a fundamental psychological need essential for wellbeing [62]. The promotion of autonomy may empower individuals to make meaningful life choices, enhancing life satisfaction and psychological health [63]. These findings also corroborate previous research on the benefits of social relations. For instance, the improvement in social resources aligns with the social convoy model, which emphasizes the importance of social networks in providing support throughout life [64].

In terms of social relationships, the training group had higher levels of social resources and a greater desire to improve relationships, both of which are key determinants of wellbeing [9]. Earlier studies highlighted the role of social integration and support in enhancing individual wellbeing and health outcomes [65].

In contrast, the control group showed a strong connection between sense of coherence and meaning in life, with aspirations for personal wellbeing and relationships being most pronounced. Sense of coherence involves perceiving life as comprehensible, manageable, and meaningful [66]. While the control group aspired to improve wellbeing and relationships, they may lack the tools or interventions to translate these aspirations into improved outcomes.

The third aim was to determine the most influential variables within these networks and analyze how their impact varies between participants who have taken the training and those who have not.

By analyzing the network’s influential variables, training status was found as the central connector between sociodemographic factors and training aim variables. In the unified network, it was directly and positively linked to key psychosocial factors, including transcendence, aspirations for societal impact and personal growth, aspirations for pursuit of fame and wealth, social resources, aspirations for personal wellbeing and relationships, and desire to improve quality of relationships. This suggests that participation in the training program not only relates to certain demographic characteristics but also specific psychosocial aspirations and resources.

The training status is related to both intrinsic and extrinsic aspirations. The positive association between training status and aspirations for societal impact and personal growth implies that training participants report higher contributions to society and pursuit of personal development. This aligns with the Self-Determination Theory, which emphasizes the importance of intrinsic motivation and personal growth in wellbeing [62]. Similarly, the link to aspirations for the pursuit of fame and wealth shows higher levels of extrinsic goals in the training group, acting as motivators impacting personal outcomes [67]. This study is one of the few to address the idea of the symbiotic effect of fostering intrinsic and extrinsic goals [68].

Training status was also positively associated with social resources and the desire to improve quality of relationships. This finding emphasizes the importance of social support networks in personal development and psychological wellbeing, offering practical implications for designing interventions. Strengthening social resources can lead to better health outcomes, as social support is a key determinant of wellbeing [9,12].

Transcendence, involving spiritual growth or a sense of connectedness beyond the self, was also positively and directly linked to training status and also more pronounced in the training group, meaning the training participants seemed to strive for deeper more meaningful connections with themselves, others, and the world around them [63,69]. This is also consistent with studies indicating that transcendental experiences contribute to psychological wellbeing and life satisfaction [70,71].

However, it is important to interpret these associations with caution. Personal characteristics can change in the course of life or through life events but are often considered relatively stable and may not change easily or quickly through interventions of limited duration [72]. The observed differences between the training and control groups may therefore reflect pre-existing differences rather than changes caused by the training program. Individuals who choose to participate in the training programs may already possess certain characteristics or motivations that distinguish them from those who do not aim to participate [73].

The findings of this study suggest an association between participation in the training program and certain psychosocial variables, but the cross-sectional and non-randomized design of the study limits the ability to determine whether the training program directly influences these personal characteristics.

### 4.1. Strengths of the Study

The standardized training program was grounded in Lozanov’s superlearning and positive psychology principles, incorporating a multidisciplinary combination of theoretical knowledge and practical exercises designed to target a wide range of personal and social resources. This ensured a broad spectrum of skills development and replicable results.

Employing network analysis and network comparison techniques in psychosocial research is quite new and offers a detailed exploration of the interdependencies between demographic factors and various psychological and social dimensions, providing a richer understanding of the training’s impacts.

The use of elastic-net regularized neighborhood regression in network analysis allowed for the control of confounding variables and enhanced the robustness of the study’s findings, despite the inherent complexities of the dataset.

The training program’s intensive nature, comprising 21 days of in-person training, combined with high participant engagement—in group sessions, one-on-one sessions, and peer group meetings—and extended duration over a full year provides enough time for deep learning and the application of new skills. This extended program duration was designed to provide participants with opportunities to practice and apply learned strategies in their daily lives between modules.

### 4.2. Limitations of the Study

Without random assignment to the training and control group, selection biases may have been introduced, as participants self-selected into the training program. This could influence the outcomes due to pre-existing differences between the groups.

The cross-sectional design limited the ability to draw conclusions about the causality effects, hence longitudinal studies are needed in future research. Due to this design, the time elapsed since finishing the training varied amongst the training participants (3.4 years on average since finishing the training), potentially affecting the retention and impact of the program differently across individuals.

The relatively small sample size, with 140 participants in the control group and 83 in the training group, may limit the statistical power of the study. This constraint could also affect the generalizability of the findings to a broader population, as it may not adequately represent all individuals who might participate in such training programs.

The unequal sample sizes between the training group (83 participants) and the control group (140 participants) may limit the reliability of the comparisons. This imbalance should be considered when interpreting the results.

The research was confined to German-speaking populations in specific European countries, which might restrict the generalizability of the findings to other populations in different cultural, social, or economic contexts.

Relying primarily on self-reported measures can introduce biases such as social desirability or recall bias, which may affect the accuracy of the data regarding participants’ behaviors and experiences.

Personal characteristics are often considered relatively stable over time and may require long-term multifaceted interventions to observe significant changes. The training program under research may fulfill these requirements but the study did not control for baseline levels of these characteristics, making it difficult to determine whether observed differences are attributable exclusively to the training program.

The financial cost of the training program potentially limits access for those who cannot afford such investments. This economic barrier excludes individuals from lower socio-economic backgrounds who may benefit from the training but are unable to participate due to cost constraints. On the other hand, higher education and income levels in the training group may provide advantages that contribute to wellbeing independently of the training.

### 4.3. Implications for Practice

The study shows the importance of tailored interventions and incorporating comprehensive and training modules that address various aspects of human development to enhance quality of life.

The findings on sense of coherence suggest that training programs should include elements that help individuals perceive their world as more comprehensible, manageable, and meaningful. Given the positive impact of improved social relationships on overall wellbeing, training programs should also focus on developing these competencies. Continuous support and exposure to the program through follow-up sessions may help solidify new skills and ensure the sustainability of behavioral changes.

The findings can inform policymakers about the benefits and design of adult education programs focused on psychosocial competencies. Insights into the interaction between training and improved psychosocial outcomes can guide clinicians and counselors in recommending or designing similar interventions. The findings underline the importance of making such training accessible to a broader audience, including those who may be economically disadvantaged.

### 4.4. Future Research Directions

Implementing a longitudinal design would help ascertain the causality of interactions and training benefits on Life Satisfaction and psychological wellbeing. Including participants from different cultural backgrounds and countries could enhance the validity of the findings. To better control for selection biases and improve the reliability of the outcomes, future studies should aim to use RCTs where feasible.

## 5. Conclusions

In summary, our study successfully achieved its aims and found significant differences in the network structure of the training group compared to the control group and that participants in the holistic training program showed higher levels of psychosocial wellbeing. Training status was found to be the most central and influential variable, directly linked to aspirations for societal impact and personal growth, pursuit of fame and wealth, transcendence, desire to improve the quality of relationships, aspirations for personal wellbeing and relationships, and the presence of social resources. The training group showed stronger connections between wellbeing, Life Satisfaction, meaning in life, and transcendence, emphasizing autonomy and private social relationships. On the one hand, this aligns with Seligman’s work on positive psychology interventions that enhance wellbeing through cultivating positive emotions, engagement, and meaning [25] and on the other hand with the principles of the Self-Determination Theory, which emphasizes the role of autonomy, competence, and relatedness in strengthening intrinsic motivation and mental health [65]. In contrast, the control group’s network primarily connected through the “Sense of Coherence–Meaning in Life” axis, with pronounced aspirations for personal wellbeing and relationships, aligning with Antonovsky’s concept on Sense Of Coherence, that suggests that comprehensibility, manageability, and meaningfulness are important for coping with stress and maintaining health [64]. The employment of advanced analytical methods, such as network analysis [49] and network comparison tests, provides an innovative contribution to understanding the applicability of these methods in psychosocial research. However, methodological limitations—including the cross-sectional, non-randomized design, group differences, and potential selection biases—complicate drawing conclusions on the effect of the training program and need to be made with caution.

From a public health perspective, this study offers valuable insights into how holistic training programs can positively influence personal growth and psychosocial outcomes. The findings, while still preliminary and subject to limitations, provide initial indications that such programs could potentially contribute to individuals’ aspirations and resources, possibly influencing life satisfaction and psychological wellbeing. By identifying the central role of training status and its impact on various psychosocial factors, our study underscores the importance of further investigating the role of comprehensive interventions that address multiple dimensions of personal development.

The training under research also introduces the application of Lozanov’s superlearning [24] approach to psychosocial research. It was originally developed to optimize learning and retention and has primarily been researched in language teaching. Therefore, this work presents an early exploration of superlearning principles in the psychosocial domain.

Future studies are required to identify the underlying mechanisms through which holistic training programs affect psychosocial wellbeing. Longitudinal designs, larger and more equal sample sizes, and randomized controlled trials would help assess the causality and sustainability of the training’s impact. Additionally, including participants from diverse cultural backgrounds and socioeconomic statuses could enhance the generalizability of the findings.

In conclusion, our study contributes to the existing body of knowledge by demonstrating the associations of a holistic training program with personal growth, societal impact aspirations, and transcendence. It highlights the importance of integrating multiple theoretical frameworks and employing advanced analytical methods to deepen our understanding of psychosocial wellbeing. This provides a foundation for developing effective interventions to promote individual and community wellbeing.

## Figures and Tables

**Figure 1 ijerph-21-01654-f001:**
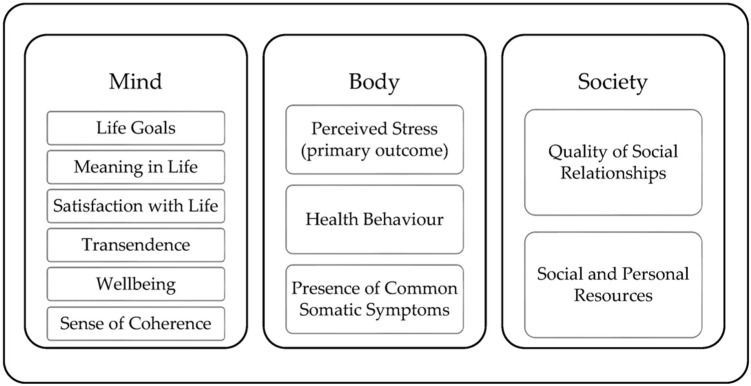
The three primary domains—mind, body, and society—covered by the standardized questionnaires and custom items.

**Figure 2 ijerph-21-01654-f002:**
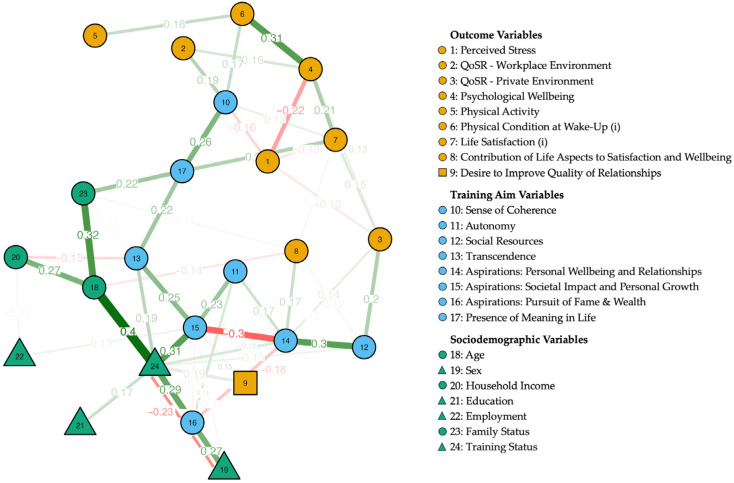
Estimated unified network structure (*n* = 223). Variables are represented as circles (continuous), squares (ordinal), and triangles (nominal) and lines represent edges (i.e., association between two variables); the thicknesses and intensity of the lines indicate the weights of the edges. Green edges indicate a positive association and red edges indicate a negative association, associations involving ordinal variables are kept in grey.

**Figure 3 ijerph-21-01654-f003:**
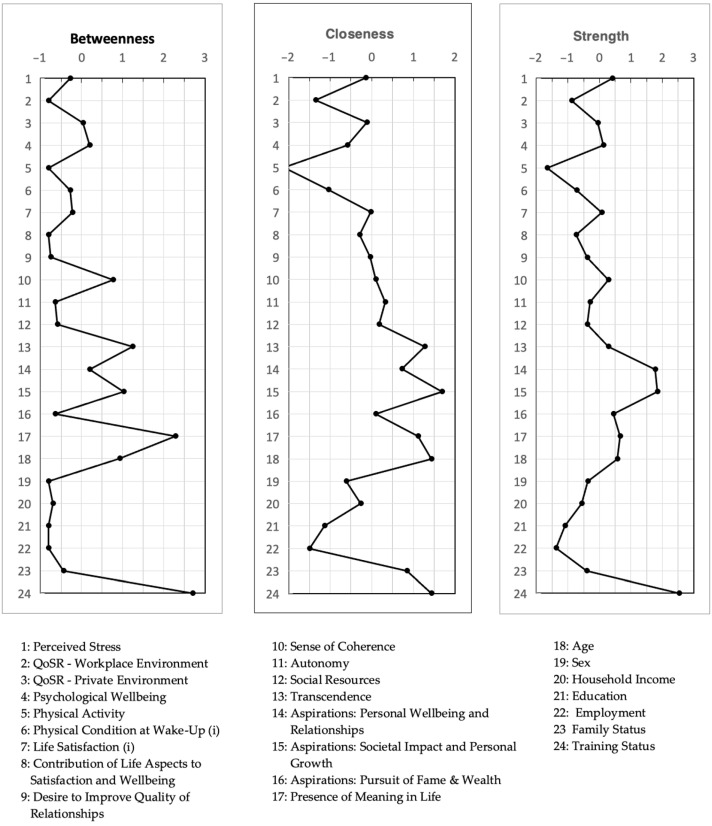
Centrality indices of the unified network. Illustrates the values of each variable for betweenness, closeness, and strength.

**Figure 4 ijerph-21-01654-f004:**
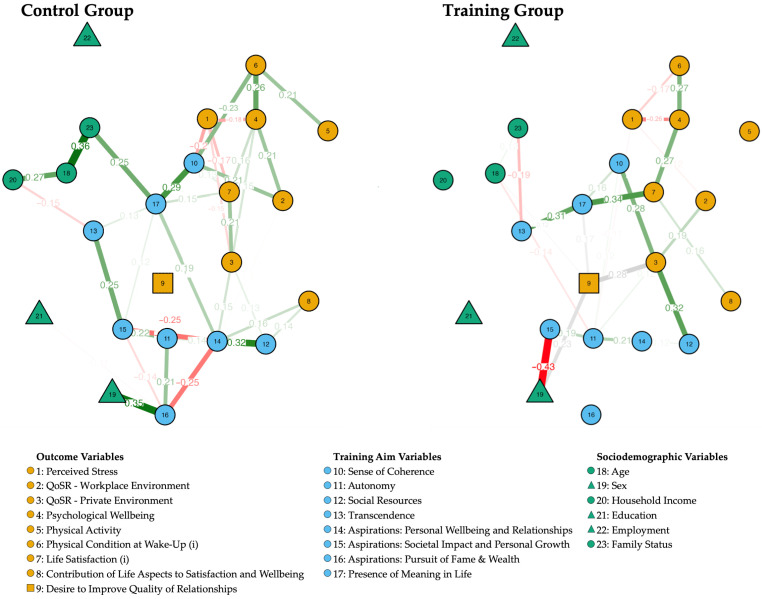
Estimated network structure for the control (*n* = 83) and training group (*n* = 140). Variables are represented as circles (continuous), squares (ordinal), and triangles (nominal) and lines represent edges (i.e., association between two variables); the thicknesses and intensity of the lines indicate the weights of the edges. Green edges indicate a positive association and red edges indicate a negative association; associations involving ordinal variables are kept in grey.

**Table 1 ijerph-21-01654-t001:** Training content overview.

Training Content
Stress and Somatic Symptom Management: Identification and management of stress triggers and common somatic symptoms like headaches, fatigue, and gastrointestinal issues. Participants engaged in biofeedback sessions, physical relaxation exercises, and learned about stress and psychosomatic connections.
Life Satisfaction and Positive Psychology: Enhancing life satisfaction and understanding the power of positive thoughts. Activities included gratitude practices, goal setting, journaling, vision board creation, and group discussions on personal strengths and achievements.
Social Relationships and Community Building: Improving the quality of social relationships and strengthening the sense of belonging. Activities included social skills training and group activities to build trust and practice communication skills.
Health Behavior and Lifestyle: Enabling participants to understand their true health needs and empowering them to modify habits through meditation practices and complementary strategies.
Overall Wellbeing and Personal Development: Providing a comprehensive approach, integrating physical, mental, and emotional health. Activities included personalized development plans, and workshops on personal growth and self-actualization.

**Table 2 ijerph-21-01654-t002:** Full list of instruments used in the study.

Domain	Tool
Health behavior	Evaluated using the FEG—Questionnaire for the Assessment of Health Behavior. Covers the behaviors that are relevant to this study: exercise, sleep, and psychological stress [27].
Life goals	Assessed by the Aspiration Index that evaluates internal and extrinsic ambitions, which can be viewed as life goals [28].
Meaning in life	Assessed by the MLQ—Meaning in Life Questionnaire, which distinguishes between presence of meaning and searching for meaning [29].
Perceived stress	Assessed by the PSS-10—Perceived Stress Scale [30].
Personal and social skills	Personal, societal, and structural resources are assessed by the ERI—Essener Ressourceninventar [31].
Presence of common somatic symptoms	Assessed by the FEG—Questionnaire for the Assessment of Health Behavior [27].
Quality of Social Relationships (QoSR)	Assessed by EVOS—Evaluation of Social Systems and EXIS—Experience of Social Systems questionnaires for private and workplace environments. [32,33].
Satisfaction with Life	Assessed by the LS-4—Life Satisfaction questionnaire [34].
Sense of Coherence	Assessed by the SOC-L9—Sense of Coherence Scale [35].
Transcendence	Assessed by the CSRF—Character Strength and Rating Form (Transcendence part only) [36].
Psychological Wellbeing	Assessed by the WHO-5 (by World Health Organization) questionnaire for Psychological Wellbeing [37].

**Table 3 ijerph-21-01654-t003:** Descriptive statistics and primary analysis results.

Factor	Control Group (*n* = 144)	Training Group (*n* = 83)	*p*-Value
Age (years)	41 (92.99)	50 (144.06)	<0.001
Sex			<0.001
Male	63 (45.0%)	18 (21.7%)	
Female	75 (53.6%)	65 (78.3%)	
Other	2 (1.4%)	0 (0.0%)	
Education			0.025
Still in school	2 (1.6%)	0 (0.0%)	
High school	4 (3.1%)	0 (0.0%)	
Secondary school	56 (43.4%)	19 (27.9%)	
Academy	9 (7.0%)	11 (16.2%)	
University/College	53 (41.1%)	36 (52.9%)	
Other	5 (3.9%)	2 (2.9%)	
Household Income			0.012
<Median	31 (30.4%)	9 (14.1%)	
≥Median	71 (69.6%)	55 (85.9%)	
Individual Income			0.436
<Median	49 (42.6%)	29 (40.3%)	
≥Median	66 (57.4%)	43 (59.7%)	
Employment			<0.001
Full-time	69 (49.3%)	28 (33.4%)	
Part-time	21 (15.0%)	12 (14.5%)	
Minor-employed	2 (1.4%)	0 (0.0%)	
Retired	11 (7.9%)	3 (3.6%)	
Self-employed	16 (11.4%)	36 (43.4%)	
In training	16 (11.4%)	2 (2.4%)	
Parental leave	2 (1.4%)	1 (1.2%)	
Duration Employment	8 (112.17)	8 (111.72)	0.960
Living Environment			0.123
City/Urban	57 (40.7%)	41 (49.4%)	
Town	27 (19.3%)	20 (24.1%)	
Rural	56 (40.0%)	22 (26.5%)	
Household Composition			0.744
Alone	18 (12.9%)	13 (15.7%)	
In shared apartment	11 (4.9%)	8 (5.7%)	
With family	102 (72.9%)	62 (74.7%)	
With parents	17 (7.6%)	12 (8.6%)	
Relationship Status			0.060
Single	12 (8.7%)	5 (6.0%)	
In a relationship	59 (42.8%)	24 (28.9%)	
Married/Partnered	63 (45.7%)	48 (57.8%)	
Divorced	3 (2.2%)	6 (7.2%)	
Widowed	1 (0.7%)	0 (0.0%)	
Family Status			0.016
No children	60 (42.9%)	23 (27.7%)	
With children	80 (57.1%)	60 (72.3%)	

**Table 4 ijerph-21-01654-t004:** Detailed analysis of the variables meeting the inclusion criteria.

Factor	Control Group	Training Group	*p*-Value
Perceived Stress Level	15 (122.76)	12 (93.84)	0.001
Health Behavior			
Sleep Problems	7 (115.81)	7 (105.58)	0.249
Physical Condition at Wake-Up	17 (122.37)	14 (94.51)	0.002
Contribution of Life Aspects to Satisfaction and Wellbeing	32 (105.04)	33 (123.73)	0.036
Contribution of Life Aspects to Problems and Difficulties	15.5 (117.21)	14 (103.21)	0.116
Evaluation of Problems	2 (107.82)	3 (119.05)	0.206
Physical Activity	14 (104.52)	15 (124.61)	0.024
Physical Activity through negative emotions	9 (111.09)	9 (113.54)	0.783
Medication	2 (115.91)	2 (105.4)	0.223
Presence of Common Somatic Symptoms	11 (111.74)	11 (112.43)	0.938
Satisfaction with Life	7 (120.52)	7 (97.63)	0.010
Quality of Social Relationships (QoSR)			
Workplace Environment	4.31 (100.95)	4.73 (130.64)	<0.001
Private Environment	4.42 (104.21)	4.7 (125.13)	0.019
Desire to Improve Quality of Relationships			0.039
No hope for improvement	0 (0%)	1 (1.2%)	
Generally, yes	13 (9.3%)	16 (19.3%)	
Depends on the person	63 (45.0%)	27 (32.5%)	
Satisfied as it is	64 (62.1%)	39 (37.9%)	
Psychological Wellbeing	58 (101.41)	68 (129.86)	0.001
Aspirations			
Personal Wellbeing and Relationships	−0.052 (101.76)	0.313 (129.27)	0.002
Pursuit of Fame and Wealth	−0.012 ± 0.994	0.210 ± 0.980	0.015
Valuation of Fame and Wealth	0.009 ± 1.017	−0.016 ± 0.977	0.856
Societal Impact and Personal Growth	−0.126 (94.86)	0.474 (140.9)	<0.001
Meaning in Life			
Presence of Meaning	28 (101.25)	31 (130.13)	0.001
Search of Meaning	19.5 (109.9)	20 (115.54)	0.528
Sense of Coherence	5.56 (101.51)	5.89 (129.69)	0.002
Inventory of Resources			
Social Resources	3.6 (100.95)	3.8 (130.64)	<0.001
Autonomy	3.25 (103.68)	3.5 (126.04)	0.011
Internal Locus of Control	3 (106.82)	3 (120.73)	0.113
Transcendence	6.4 (96.18)	7.4 (138.68)	<0.001

**Table 5 ijerph-21-01654-t005:** Main results of the unified network diagram analysis.

Inter-Variable-Connections	Edge Weight
Strongest Connections	
Training Status ↔ Age	0.401
Training Status ↔ Societal Impact and Personal Growth	0.307
Physical Condition at Wake-Up (i) ↔ Psychological Wellbeing	0.307
Social Resources ↔ Personal Wellbeing and Relationships	0.304
Societal Impact and Personal Growth ↔ Personal Wellbeing and Relationships	−0.298
Training Status ↔ Pursuit of Fame and Wealth	0.291
Sex ↔ Pursuit of Fame and Wealth	0.270
Age ↔ Household Income	0.266
Age ↔ Family Status	0.266
Sense of Coherence ↔ Presence of Meaning in Life	0.260
Transcendence ↔ Societal Impact and Personal Growth	0.246
Training Status ↔ Sex	−0.231
Autonomy ↔ Societal Impact and Personal Growth	0.226
Transcendence ↔ Presence of Meaning in Life	0.221
Perceived Stress ↔ Psychological Wellbeing	−0.220
Psychological Wellbeing ↔ Life Satisfaction (i)	0.207
QoSR—Private Environment ↔ Social Resources	0.204
Life Satisfaction (i) ↔ Presence of Meaning in Life	0.200
Connections Between Training Aims and Outcome Variables	
Sense of Coherence	
QoSR—Workplace Environment	0.194
Physical Condition at Wake-Up (i)	0.173
Perceived Stress	−0.156
Life Satisfaction (i)	0.133
Social Resources	
QoSR—Private Environment	0.204
Contributions of Life Aspects to Satisfaction and Wellbeing	0.115
Personal Wellbeing and Relationships	
QoSR—Private Environment	0.136
Contributions of Life Aspects to Satisfaction and Wellbeing	0.171
Societal Impact and Personal Growth	
Desire to Improve Quality of Relationships	0.147
Presence of Meaning in Life	
Life Satisfaction (i)	0.214
Connections with Training Status Variable	
Age	0.401
Societal Impact and Personal Growth	0.307
Pursuit of Fame and Wealth	0.291
Sex	−0.231
Transcendence	0.188
Desire to Improve Quality of Relationships	0.186
Education	0.171
Personal Wellbeing and Relationships	0.149
Social Resources	0.125

**Table 6 ijerph-21-01654-t006:** Main results of the network diagram analysis for the control and training group.

Inter-Variable-Connections	Edge Weight
Control Group	
Age ↔ Family Status	0.358
Aspirations: Pursuit of Fame and Wealth ↔ Sex	0.347
Social Resources ↔ Aspirations: Personal Wellbeing and Relationships	0.320
Sense of Coherence ↔ Presence of Meaning in Life	0.293
Age ↔ Household Income	0.267
Psychological Wellbeing ↔ Physical Condition at Wake-Up (i)	0.264
Aspirations: Personal Wellbeing and Relationships ↔ Aspirations: Societal Impact and Personal Growth	−0.252
Transcendence ↔ Aspirations: Societal Impact and Personal Growth	0.251
QoSR—Workplace Environment ↔ Sense of Coherence	0.222
Aspirations: Personal Wellbeing and Relationships ↔ Aspirations: Pursuit of Fame and Wealth	−0.246
Physical Condition at Wake-Up (i) ↔ Sense of Coherence	0.227
Training Group	
Aspirations: Societal Impact and Personal Growth ↔ Sex	−0.427
Life Satisfaction (i) ↔ Presence of Meaning in Life	0.335
QoSR—Private Environment ↔ Social Resources	0.319
Transcendence ↔ Presence of Meaning in Life	0.311
QoSR—Private Environment ↔ Desire to Improve Quality of Relationships	0.283
QoSR—Private Environment ↔ Sense of Coherence	0.275
Psychological Wellbeing ↔ Life Satisfaction (i)	0.274
Psychological Wellbeing ↔ Physical Condition at Wake-Up (i)	0.273
Perceived Stress ↔ Psychological Wellbeing	−0.261
Desire to Improve Quality of Relationships ↔ Sex	0.233

**Table 7 ijerph-21-01654-t007:** Number of connections for variables in each group.

#	Variable
Control Group	
8	Aspirations: Personal Wellbeing and Relationships
7	QoSR—Private Environment
6	Presence of Meaning in Life
6	Aspirations: Societal Impact and Personal Growth
5	Sense of Coherence
5	Aspirations: Pursuit of Fame and Wealth
5	Psychological Wellbeing
5	Life Satisfaction (i)
5	Perceived Stress
Training Group	
6	Autonomy
4	QoSR—Private Environment
4	Presence of Meaning in Life
4	Life Satisfaction (i)
4	Desire to Improve Quality of Relationships
4	Sense of Coherence
4	Perceived Stress
3	Psychological Wellbeing
3	QoSR—Workplace Environment
3	Age

**#**: The variables are ranked based on the number (#) of connections they have with absolute edge weights greater than 0.1.

## Data Availability

Data are contained within the article and Appendix A.

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
