# Peer review of "The Relationship Between Body, Mind, and Social Characteristics in a Sample of a Training Program for Developing Social and Personal Resources: A Network Analysis"

_ijerph, 2024, doi:10.3390/ijerph21121654_

Round 1

Reviewer 1 Report

Comments and Suggestions for Authors

Overall, this is a very interesting study.  The introduction sufficiently justified the need for the research.  The paper is well written and well presented.

Methods are described and results are presented in detail.

There are some points in the methods that need clarified.  For instance, in the Participants section in the methods, what do you mean "requiring half the expected time to finish survey or answering control questions incorrectly".  Why exactly is this a problem, what did this tell you that meant they had to be excluded?  What control questions did you ask - what was a correct answer and what was an incorrect answer - please clarify.  

In section 3.2.1 - what do you mean "met the criteria"?  What criteria?  Please clarify.

In section 3.2.2. - "these criteria" - what criteria is this referring to - please clarify.

Although the authors acknowledge that the non-randomised design was a limitation, they could have more strongly discussed the implications of this for their results.  For instance, the training group and control group were quite different from each other so it is not convincing that the training programme was the active ingredient that led to the outcomes.  

Author Response

We kindly thank the reviewer for the comments and valuable suggestions that provided important information for the revision of the manuscript and helped improve it significantly!

Comment 1: There are some points in the methods that need clarified.  For instance, in the Participants section in the methods, what do you mean "requiring half the expected time to finish survey or answering control questions incorrectly".  Why exactly is this a problem, what did this tell you that meant they had to be excluded?  What control questions did you ask - what was a correct answer and what was an incorrect answer - please clarify.  ]
Response 1: Thank you for pointing this out, this is a very valid question and we are happy to clarify: In the test runs to fill out the questionnaires the average time needed to finish was 40minutes. To avoid having data from participants that only click through (and giving random answers) without reading, we set the time-limit to 20minutes (as there are people that might read and answer significantly faster than we did). We did not exclude any participants due to the time-limit. To increase the quality of the answers given (the questions have been properly read) three control questions have been implemented: "Please do not select any of the answers below and click continue" (this question included an explanation that it is a control question to make sure that participants are focused), "Please choose the answer "green" and click continue", "Please choose answer "8" and continue". When answering any of the control questions incorrectly, the questionnaire ended. This was explained to the study participants in the introduction of the survey. Participants were able to start the survey again in case they were screened out. The screen out page also contained the information that the wrong answer was given in one of the control questions.

Comment 2: In section 3.2.1 - what do you mean "met the criteria"?  What criteria?  Please clarify. In section 3.2.2. - "these criteria" - what criteria is this referring to - please clarify.
Response 2: Thank you for pointing this out, in fact it is not described well enough. "The criteria" refer to the criteria described in "2.5. Statistical Analysis". Accordingly, I have modified the text, that now says: "The following variables met the criteria described in section 2.5. and were included in the network analysis..." (line 264), "The following variables met the criteria described in section 2.5. and were included..." (line 270)

Comment 3: Although the authors acknowledge that the non-randomised design was a limitation, they could have more strongly discussed the implications of this for their results.  For instance, the training group and control group were quite different from each other so it is not convincing that the training programme was the active ingredient that led to the outcomes.  
Response 3: Yes, we totally agree, the non-randomization as well as the group differences (especially in demographic variables) are a limitation. To emphasize this point I added this concern in the discussion section: "However, it is important to interpret these associations with caution. Personal characteristics can change in the course of life or through life events but are often considered relatively stable and may not change easily or quickly through interventions of limited duration []. The observed differences between the training and control groups therefore may reflect pre-existing differences rather than changes caused by the training program. Individuals who choose to participate in the training programs may already possess certain characteristics or motivations that distinguish them from those who do not aim to participate []." (Line 475), in the limitations section: "The unequal sample sizes between the training group (83 participants) and the control group (140 participants) may limit the reliability of the comparisons. This imbalance should be considered when interpreting the results." (Line 521) and in the conclusion section: "Longitudinal designs, larger and more equal sample sizes and randomized controlled trials would help assess causality and sustainability of the training's impact." (Line 605)

Reviewer 2 Report

Comments and Suggestions for Authors

This is a well written paper utilising network analysis to analyse connections between potential training programme benefits and demographic information. I have noted the following minor points for clarification to be reviewed.

General concept comments:

- Article: The paper is well written and has a clear structure. The tables and presentation of graphs is good. The authors set a number of aims: investigate the interactions between demographic factors and the body/mind/social characteristics of participants within the training programme; compare network structures of training and control groups; determine the most influential variables within these networks and analyse how their impact varies between trained and non-trained participants. There is detailed discussion about these aspects, however, it would be helpful for the reader if the authors could specify to what extent they met these aims in the paper. As it stands there is no mention of aims in the discussion nor conclusion even though contribution to the field is discussed.

- Review: As I have raised below, I feel the relationship of the authors to the training programme is unclear. What was the motivation for this piece of research and why was this specific training programme selected? How did the authors gain access to the data? The methods of the research appear robust but the sample size is small. The sample size only just exceeds that of the minimal expectation for a survey study and it is positive that this has been discussed in the limitations section. The control group is also substantially greater than the training group which is slightly concerning as equal group sizes would leave the sample below N 200. Would it be possible to extend this study to increase the sample size?

Specific comments:

- Please check the abstract for inconsistent use of capital letters (18, 27) and use of full comma (17).

-          Good introduction outlining why mental health is a public health concern and holistic interventions are required. There could be more mention of training programmes considering the focus on the paper.

-          Please could you revise this section as it is slightly unclear how the non-randomization related to the self-selection process of the training programme (lines 96-99).

-          Are you able to say what the split of participants was between the three countries? (line127). If this was not considered relevant for the study, what was the rationale for this?

-          Check spelling line 395

-          What are the statements about intensivity and engagement levels of the training based on? Please can this be clarified (lines 483-485).

-          The statement about gradual integration of behaviours is somewhat sceptical. Please amend or consider removing this (lines 485-487).

-          Check grammar (lines 490-491).

-          The statement about existing body of knowledge could benefit from references: what is this body of knowledge and who are the key authors' whose work you are contributing to? (Line 560)

- The authors declare no conflict of interest nor funding for this piece of work. It would be helpful to understand who commissioned this research and how the authors were able to access the data. If they were involved in delivering the training, this should be clarified in the paper.

Author Response

We kindly thank the reviewer for the comments and valuable suggestions that provided important information for the revision of the manuscript and helped improve it significantly!

Comment 1: - Article: The paper is well written and has a clear structure. The tables and presentation of graphs is good. The authors set a number of aims: investigate the interactions between demographic factors and the body/mind/social characteristics of participants within the training programme; compare network structures of training and control groups; determine the most influential variables within these networks and analyse how their impact varies between trained and non-trained participants. There is detailed discussion about these aspects, however, it would be helpful for the reader if the authors could specify to what extent they met these aims in the paper. As it stands there is no mention of aims in the discussion nor conclusion even though contribution to the field is discussed.
Response 1: Thank you for pointing this out. This comment brings a great value to the publication and we are great you brought this up. The discussion section was restructured and is now sectioned according to the three aims, which gives more structure and clarity to the text. (Line 395)

Comment 2: - Review: As I have raised below, I feel the relationship of the authors to the training programme is unclear. What was the motivation for this piece of research and why was this specific training programme selected? How did the authors gain access to the data?
Response 2: These are very important questions and we hope we have answered them to the reviewer's satisfaction: This research was initiated by the first author, Christoph Janka, who participated in the training in 2011. The training program fulfilled the first author's criteria for a holistic training program (incorporating body, mind, and social aspects), uses superlearning principles in their teaching (previous studies have mainly focused on superlearning in language teaching), stands out when it comes to duration and participant engagement compared to other trainings in the literature and is unique in the german speaking countries. The motivation was to not only explore the relations of these holistic aspects, but also investigate if an intervention can improve any of these and act as an early health promotion intervention in the general population, and, most importantly, contribute to science by studying innovative concepts. Institute Kutschera, the training provider, agreed to support this research by inviting their alumni to participate in this study and providing information on the training content. None of the authors were involved in designing or delivering any components of the training program, nor did they receive any compensation. Christoph Janka’s contact with the training provider in 2023 was limited to organizational matters related to recruitment timing and duration.

The following text was added to the conflict of interest section: "C.J. is familiar with the training programme in this study as a participant himself in 2011. The contact to Institut Kutschera GmbH in 2019 and 2023 was strictly limited to administrations related to this research to avoid any conflict of interest. Therefore, the authors declare no conflict of interest." (Line 639)

Comment 3: The methods of the research appear robust but the sample size is small. The sample size only just exceeds that of the minimal expectation for a survey study and it is positive that this has been discussed in the limitations section. The control group is also substantially greater than the training group which is slightly concerning as equal group sizes would leave the sample below N 200. Would it be possible to extend this study to increase the sample size?
Response 3: Thank you for pointing out this deficiency regarding the sample size limitations. We acknowledge that the relatively small sample size and unequal group sizes are key limitations of this study, which warrants caution in interpretation. We have emphasized these limitations in the revised manuscript. Unfortunately the recruiting phase was limited to a 2 months window in 2023, during which response rates declined significantly in the final weeks despite active recruitment efforts. In the future research directions section we pointed out, that besides using a longitudinal, randomized control design, a larger, more balances sample size are recommended.

Comment 4: - Please check the abstract for inconsistent use of capital letters (18, 27) and use of full comma (17).
Response 4: Thank you for pointing this out, the text was corrected accordingly.

Comment 5: Good introduction outlining why mental health is a public health concern and holistic interventions are required. There could be more mention of training programmes considering the focus on the paper.
Response 5: Thank you for this comment, we have updated the introduction section accordingly: "For example, mindfulness-based stress reduction programs have been shown to reduce symptoms of anxiety and depression in both clinical and non-clinical populations [[]. Cognitive-behavioral training interventions have also demonstrated efficacy in enhancing coping strategies and reducing stress levels []." (Line 59). "In response, holistic training programs that simultaneously address personal development and social integration to promote wellbeing are getting more into focus []. Previous studies have highlighted the potential of holistic interventions in improving mental health outcomes, enhancing life satisfaction, and strengthening social bonds []." (Line 74)

Comment 6: Please could you revise this section as it is slightly unclear how the non-randomization related to the self-selection process of the training programme (lines 96-99)
Response 6: Thank you very much for your input. We agree that the text is misleading as randomization in this design is not possible and therefore the term "non-randomization" is obsolete and has been removed from the text. Training alumni have been recruited through the training provider (the programme available on the market) and the control group was recruited in the general population, both being assessed once and at the same time - the training group participants had already participated in the training (and back then voluntarily decided to take the training) before recruiting started .
We therefore modified the text, hoping to bring more clarity: "This study employed a cross-sectional design to investigate the interaction between demographics and body, mind, and social characteristics in participants of a training program aimed at developing social and personal resources. The reason for the cross-sectional design was due to practical constraints - training participants had previously taken and finished the training, making a longitudinal design with baseline measurements infeasible." (Line 102).

Comment 7:  Are you able to say what the split of participants was between the three countries? (line127). If this was not considered relevant for the study, what was the rationale for this?
Response 7: Thank you for this comment, adding this information brings significant value to the work. We added the number of participants for each country: "Participants were recruited from Austria (n=191), Germany (n=26), and Switzerland (n=6)." (Line 243)

Comment 8: Check spelling line 395
Response 8: Thank you for this input - the text has been corrected.

Comment 9: What are the statements about intensivity and engagement levels of the training based on? Please can this be clarified (lines 483-485).
Response 9: Thank you for pointing this out. We agree with this comment as it was not clear what these terms mean in relation to the training. We have therefore modified the text, hoping to bring the expected clarity: "The training program’s intensive nature – comprising 21 days of in-person training -  combined with high participant engagement - in group sessions, one-on-one sessions and peer group meetings - and extended duration over a full year provides enough time for deep learning and the application of new skills." (Line 501)

Comment 10: The statement about gradual integration of behaviours is somewhat sceptical. Please amend or consider removing this (lines 485-487).
Response 10: Thank you for pointing this out. We have amended the text to focus more on the program structure and avoid assumptions and claims: "This extended program duration was designed to provide participants with opportunities to practice and apply learned strategies in their daily lives between modules." (Line 504)

Comment 11: Check grammar (lines 490-491)
Response 11: Thank you very much for this input, the text has been amended: "Without random assignment to the training and control group, selection biases may have been introduced, as participants self-selected into the training program." (Line 509)

Comment 12: The statement about existing body of knowledge could benefit from references: what is this body of knowledge and who are the key authors' whose work you are contributing to? (Line 560)
Response 12: Thank you for pointing this out. This is a very important aspect and it was underrepresented so far. We have now highlighted the relations to existing fields of research to which this study contributes: "On the one hand, this aligns with Seligman's work on positive psychology interventions that enhance wellbeing through cultivating positive emotions, engagement, and meaning [19] and on the other hand with the principles of the Self-Determination Theory, which emphasizes the role of autonomy, competence, and relatedness in strengthening intrinsic motivation and mental health [55]. In contrast, the control group's network primarily connected through the "Sense of Coherence–Meaning in Life" axis, with pronounced aspirations for personal wellbeing and relationships, aligning with Antonovsky's concept on sense of coherence, that suggests comprehensibility, manageability, and meaningfulness being important for coping with stress and maintaining health [65]. The employment of advanced analytical methods, such as network analysis [43] and network comparison test provides an innovative contribution to understanding the applicability of these methods in in psychosocial research. " (Line 575)

"The training under research also introduces the application of Lozanov's superlearning [18] approach to psychosocial research. It was originally developed to optimize learning and retention and has primarily been researched in language teaching. Therefore, this work presents an early exploration of superlearning principles in the psychosocial domain." (Line 599)

Comment 13: The authors declare no conflict of interest nor funding for this piece of work. It would be helpful to understand who commissioned this research and how the authors were able to access the data. If they were involved in delivering the training, this should be clarified in the paper.
Response 13: Thank you for pointing this out. We tried to cover all aspects of this question in comment 2.

Reviewer 3 Report

Comments and Suggestions for Authors

Dear Author(s),

Thank you for your effort so far in getting your manuscript to review. The manuscript entitled “The Relationship between Body, Mind and Social Characteristics in a Sample of a Training Programme for Developing Social and Personal Resources: A Network Analysis”. I have shared the strengths and improvable aspects of this research below.

Strengths:

1. Network comparison analyses were used in the study.

2. The introduction is fluent.

3. It is interdisciplinary.

4. Visual analyses are given enough space.

5. Conclusion and recommendations are sufficient. References are strong.

Weaknesses or questions to be answered:

1. Ethics committee permission was obtained in 2019. The research was conducted in 2023 or later. The date range in which the data were collected in the research should be specified.

2. The differences in the number of samples between the experimental and control groups is an important deficiency.

3. It should be clearly written why samples were taken from Austria, Germany and Switzerland. The number of participants from each country should be stated separately for the experimental and control groups.

4. How long was the training programme implemented and how long after the end of the training programme was the data collected? I think a pre-test study would have been better.

5. If the criterion for selecting the sample group is to speak German well, why was data collected only from Germany?

6. Since the sample size was low, the trainings could have been given face to face.

7. The introduction focuses on mental health. The impact of the training programme on people should be mentioned. More space could be given to the topics in the content of the training programme.

Consequently, the subject of the study is important and can contribute to the literature. I find studies conducted with participants from various countries valuable. I see significant deficiencies in the methodology and I think there are questions that need to be answered. It is difficult to suggest that a training programme changes personal characteristics that are relatively more permanent. Therefore, judgements need to be made more sensitively. If the links are to be explained by demographic data, a larger sample size is needed. In conclusion, I believe that the limitations of the methodology part of the research should be strengthened with the literature. I suggest a major revision.

Author Response

We kindly thank the reviewer for the comments and valuable suggestions that provided important information for the revision of the manuscript and helped improve it significantly!

Comment 1: Ethics committee permission was obtained in 2019. The research was conducted in 2023 or later. The date range in which the data were collected in the research should be specified.
Response 1: Thank you for pointing this out. We totally agree with this comment. We have amended the text to "Adults from Austria, Germany and Switzerland participated in the study, recruitment and data collection was conducted in April and May 2023. " (Line 139)

Comment 2: The differences in the number of samples between the experimental and control groups is an important deficiency.
Response 2: Thank you for pointing this out. To emphasize this deficiency we have added the following text to the limitations: "The unequal sample sizes between the training group (83 participants) and the control group (140 participants) may limit the reliability of the comparisons. This imbalance should be considered when interpreting the results.” (Line 521)

Comment 3: It should be clearly written why samples were taken from Austria, Germany and Switzerland. The number of participants from each country should be stated separately for the experimental and control groups.
Response 3: This is a very valuable input, thank you very much. The text was amended and now says: "Adults from Austria, Germany and Switzerland - the training provider’s target countries - participated in the study, recruitment and data collection was conducted in April and May 2023." (Line 138) "Participants were recruited from Austria (n=191), Germany (n=26), and Switzerland (n=6)." (Line 244)

Comment 4: How long was the training programme implemented and how long after the end of the training programme was the data collected? I think a pre-test study would have been better.
Response 4: That is a very valuable comment, thank you for that. At first a longitudinal study with a pre-post-followup design was planned. The duration of the training is 1year, the follow-up was planned to be 6months after the training and recruitment was planned to run for 1year - resulting in a data collection period of 2,5years. Unfortunately, just as we wanted to start with the recruitment and data collection the pandemic forced the training provider to cancel the trainings. Not knowing about the length and impact of the pandemic, it was decided to change the design to cross-sectional, accepting all the deficiencies that come with it. On average, participants had completed their last training 3.4 years prior the assessment. The following limitation was added: "Due to this design the time elapsed since finishing the training varied amongst the training participants (3.4years on average since finishing the training), potentially affecting retention and impact of the program differently across individuals." (Line 513)

Comment 5: If the criterion for selecting the sample group is to speak German well, why was data collected only from Germany?
Response 5: Thank you for this question. The data was collected from participants living in Germany, Austria and Switzerland (Line 138)

Comment 6: Since the sample size was low, the trainings could have been given face to face.
Response 6: Thank you for pointing this out. The trainings were held face to face and it is now clearly written in the text as well: "The trainings were conducted in a face-to-face setting in Austria, Germany, and Switzerland." (Line 123)

Comment 7: The introduction focuses on mental health. The impact of the training programme on people should be mentioned. More space could be given to the topics in the content of the training programme.
Response 7: Thank you very much for pointing this out, your input means a substantial improvement. We have amended the text to: "In this study a holistic intervention program, designed to improve participants’ mental health and resilience and strengthen social bonds, was evaluated. This 1-year program addressed key topics such as stress and somatic symptom management, life satisfaction, social relationships, health behavior, and overall wellbeing and aimed to equip participants with practical skills integrated into a comprehensive framework for daily application.” (Line 80)

Comment 8: Consequently, the subject of the study is important and can contribute to the literature. I find studies conducted with participants from various countries valuable. I see significant deficiencies in the methodology and I think there are questions that need to be answered. It is difficult to suggest that a training programme changes personal characteristics that are relatively more permanent. Therefore, judgements need to be made more sensitively. If the links are to be explained by demographic data, a larger sample size is needed. In conclusion, I believe that the limitations of the methodology part of the research should be strengthened with the literature. I suggest a major revision.
Response 8: Thank you very much for this comment. We do agree with the statements made and we will do our best to integrate them into the text. The following amendments have been made to the text - avoiding making causal assumptions:

  • "This suggests that participation in the training program not only relates to certain demographic characteristics but also specific psychosocial aspirations and resources." (Line 452)
  • "Similarly, the link to Aspirations for pursuit of fame and wealth shows higher levels of extrinsic goals in the training group, acting as motivators impacting personal outcomes" (Line 430)
  • "Transcendence, involving spiritual growth or a sense of connectedness beyond the self, was also positively and directly linked to training status and also more pronounced in the training group, meaning the training participants seemed to strive for deeper, more meaningful connections with themselves, others, and the world around them" (Line 469)
  • Removed: "These findings suggest that the training program effectively enhances participants' sense of purpose and overall wellbeing." (Line 448)
  • "Autonomy, besides social relations, was emphasized in the training group's network structure, indicating that the training group showed higher self-determination" (Line 426)
  • Removed the promotional effect of the training on autonomy: "The promotion of autonomy may empower individuals to make meaningful life choices, enhancing life satisfaction and psychological health" (Line 454)
  • "In terms of social relationships, the training group had higher levels of social resources and a greater desire to improve relationships, both key determinant of wellbeing" (Line 434)
  • Removed: "The training may provide skills or opportunities for participants to enhance their relationships, leading to more fulfilling interpersonal interactions and support networks." (Line 462)
  • Removed: "highlighting the potential effectiveness of the training program" (Line 468)
  • Added: "However, it is important to interpret these associations with caution. Personal characteristics can change in the course of life or through life events but are often considered relatively stable and may not change easily or quickly through interventions of limited duration [66]. The observed differences between the training and control groups therefore may reflect pre-existing differences rather than changes caused by the training program. Individuals who choose to participate in the training programs may already possess certain characteristics or motivations that distinguish them from those who do not aim to participate [67].
    The findings of this study suggest an association between participation in the training program and certain psychosocial variables, but the cross-sectional and non-randomized design of the study limits the ability to determine whether the training program directly influences these personal characteristics." (Line 475)
  • Added to the limitations: "Personal characteristics are often considered relatively stable over time and may require long-term, multifaceted interventions to observe significant changes. The training program under research may fulfill these requirements but the study did not control for baseline levels of these characteristics, making it difficult to determine whether observed differences are attributable exclusively to the training program." (Line 530)
  • In the conclusion section: "In summary, our study successfully achieved its aims  and found significant differences in the network structure of the training group compared to the control group and that participants in the holistic training program showed higher levels of psychosocial wellbeing" (Line 567)
  • "However, methodological limitations—including the cross-sectional, non-randomized design, group differences and potential selection biases—complicate drawing conclusions on the effect of the training program and need to be made with caution." (Line 587)
  • "The findings, while still preliminary and subject to limitations, provide initial indications that such programs could potentially contribute to individuals' aspirations and resources, possibly influencing life satisfaction and psychological wellbeing." (Line 593)
  • "Longitudinal designs, larger sample sizes and randomized controlled trials would help assess causality and sustainability of the training's impact." (Line 605)